# Seasonal Pattern of Endo-β-Mannanase Activity During Germination of *Jeffersonia dubia*, Exhibiting Morphophysiological Dormancy

**DOI:** 10.3390/plants14020251

**Published:** 2025-01-17

**Authors:** Young Hyun Kwon, Seung Youn Lee, Yong Ha Rhie

**Affiliations:** 1Agriculture and Life Science Research Institute, Kangwon National University, Chuncheon 24341, Republic of Korea; thf4452@naver.com; 2Department of Smart Horticultural Science, Andong National University, Andong 36729, Republic of Korea; 3Department of Interdisciplinary Program in Smart Agriculture, Kangwon National University, Chuncheon 24341, Republic of Korea; 4Department of Horticulture, Kangwon National University, Chuncheon 24341, Republic of Korea

**Keywords:** embryo, lateral endosperm, micropylar endosperm, *Plagiorhegma dubium*

## Abstract

Morphophysiological dormancy (MPD) is considered one of the most primitive dormancy classes among seed plants. While extensive studies have examined the occurrence of endo-β-mannanase in seeds with physiological dormancy (PD) or non-dormancy, little is known about the activity of this enzyme in seeds with MPD. This study aimed to investigate the temporal and spatial patterns of endo-β-mannanase activity during dormancy break and germination. The research focused on *Jeffersonia dubia*, a species with deep simple MPD, by monitoring its morphological and biochemical characteristics under natural field conditions. Seeds were buried in the field and exhumed monthly over a year. Key parameters measured included germination, embryo elongation, endosperm weakening, and endo-β-mannanase activity in the exhumed seeds. Scanning electron microscopy was employed to observe structural changes in the endosperm. For the first three months after burial in May, endo-β-mannanase activity was undetectable, and the underdeveloped embryo exhibited minimal elongation. Starting in September, the embryo began to grow, accompanied by increased endo-β-mannanase activity in the micropylar endosperm. Erosion of the endosperm cell wall was observed in the lateral regions surrounding the embryo, whereas the micropylar endosperm showed no obvious signs of collapse or damage. The increase in enzyme activity coincided with moderate temperatures and a corresponding increase in embryo length. During the winter months, embryo elongation ceased for 2–3 months, and enzyme activity declined. However, as germination resumed in early March, enzyme activity increased again. This was followed by micropylar endosperm rupture and the completion of germination. The seasonal pattern of endo-β-mannanase activity observed in seeds with deep simple MPD was distinct from that of seeds with PD, MD, or non-deep MPD, highlighting the unique mechanisms underlying dormancy break and germination in *J. dubia*.

## 1. Introduction

Angiosperm seeds have cell walls rich in mannan-based polymers, which serve as a carbohydrate source to support germination and seedling growth [1]. However, in mature seeds such as lettuce and tomato, the endosperm often acts as a mechanical barrier to radicle protrusion [2]. This rigid endosperm must be weakened to facilitate germination [3]. Endosperm weakening involves cell wall hydrolysis [4], with endo-β-mannanase playing a key role in degrading these cell walls [1].

In tomato seeds, endo-β-mannanase activity emerges in the micropylar endosperm prior to germination and subsequently spreads to the lateral endosperm after germination [1,5]. However, exceptions exist, where high enzyme activity does not directly correlate with germination [6,7]. Such studies have been conducted on various species, including tomato [3,7], lettuce [8], pepper (*Capsicum annum*) [9,10], tobacco [11], melon (*Cucumis melo*) [12], *Datura ferox* [13], *Avena fatua* [14], *Lepidium sativum* [15], and *Genipa americana* [16]. Notably, these species exhibit shallow dormancy, and germination typically occurs within a short period.

In contrast, the seeds of many herbaceous plants native to temperate regions of the Northern Hemisphere possess underdeveloped embryos that must elongate before germination can occur [17]. When such seeds germinate within about 30 days under appropriate conditions, they are classified as having morphological dormancy (MD). However, if germination takes longer and requires specific treatments to break physiological dormancy (PD) before, during, or after embryo elongation, the seeds are likely to have morphophysiological dormancy (MPD)—a combination of MD and PD [17,18].

MPD can be further classified into nine levels based on the temperature requirements for embryo elongation and germination, the timing of these processes, and dormancy release in response to gibberellic acid (GA_3_) treatment [17]. Seeds with deep MPD may require several months to over two years for dormancy to break and germination to occur. Such deeply dormant seeds are considered ancestral forms of seeds with MD or PD, offering insights into the evolution of seed dormancy. Seeds with MPD do not germinate even under optimal conditions for germination and sometimes require an exceptionally long dormancy-breaking period. This is considered an advanced survival strategy for the plant [17]. For example, plant species with the deep simple MPD, one of the nine MPD types, include *Panax ginseng* [19], *Sambucus canadensis* [20], and two species of the genus *Jeffersonia* [21,22], *Epimedium koreanum* [23], and some *Ilex* species from subtropical zones [24]. These plant seeds require experiencing two seasons—summer and winter—before germinating in the following spring. These plants bloom in early spring, and their seeds mature before summer. Their germination strategy might avoid competition with other plants dominant in summer, prevent exposure to dry summer conditions, and seek the most favorable season for germination, which is spring, avoiding both the harsh conditions of summer and the cold winter periods.

Despite extensive research on endo-β-mannanase in non-dormant seeds or seeds with non-deep PD, little is known about its role in MPD. This study investigates the seasonal pattern of endo-β-mannanase activity during dormancy break in *Jeffersonia dubia* seeds to deepen our understanding of enzymatic patterns in basal angiosperms. *Jeffersonia dubia* (Berberidaceae, synonym of *Plagiorhegma dubium* Maxim.) is a perennial herb native to Eastern Asia. Renowned for its attractive light purple flowers and heart-shaped leaves, *J. dubia* is cultivated as an ornamental garden plant. In addition to its aesthetic appeal, *J. dubia* contains berberine, a bioactive compound with diverse pharmacological properties [25,26,27]. In nature, the fruit of *J. dubia* ripens in early spring, and each capsule contains more than 10 seeds. The seeds are broadly oval, long, and glossy black. Interestingly, the tip of *J. dubia* seed has an elaiosome, which ants use to bite and distribute the seeds. *Jeffersonia dubia* seeds have an underdeveloped embryo at seed maturity (dispersal), and this embryo must grow inside the seed before the seeds can germinate. Germination started in the spring of the following year, and seeds emerged as seedlings soon after germination.

## 2. Results

### 2.1. Embryo Elongation and Germination in a Field Condition

The seeds of *J. dubia* had underdeveloped embryos with the heart shape of freshly matured seeds, and the average embryo-to-seed (E:S) ratio (×10^2^) was only 9.0 ± 0.2 (mean ± SE) (Figure 1 and Figure 2A). The embryo elongation began between 6 September and 6 October 2011 (Figure 1). During this period, the embryo could be observed developing into a torpedo shape (Figure 2B,C). The embryo developed into the cotyledon stage between 6 October and 6 December 2011 (Figure 2D), coinciding with mean maximum and minimum temperatures of 20 °C and 0 °C, respectively (Figure 1). The embryo elongation slowed after 6 December and continued at a reduced pace until 6 January 2012, at which point the critical E:S ratio (×10^2^) of 96.0 ± 3.0, required for germination, was reached. Despite this, the first germination was observed on 13 February 2012, with approximately 5% of seeds germinating when sown at a depth of 1 cm. During this period, the mean maximum and minimum temperatures were 5.6 °C and −0.8 °C, respectively. Peak germination occurred as temperatures began to rise in late February, and by early March 2012, 86.7% of the seeds had germinated.

### 2.2. Observation of Endo-β-Mannanase Activity

Endo-β-mannanase activity was undetectable from June to August (Figure 3). Enzyme activity first appeared in the micropylar endosperm region in September; when the length of the embryo was still small, the E:S ratio was about 0.1. (Figure 1 and Figure 3). By October, enzyme activity had increased and was observed in both the micropylar and lateral endosperm regions (Figure 3). In November, activity was higher in the lateral endosperm than in the micropylar region. From December onward, enzyme activity decreased and remained at low levels through January and February. However, in March, germinated seeds exhibited high endo-β-mannanase activity in both the micropylar and lateral endosperm regions. Tissue printing was used to visualize the spatial distribution of endo-β-mannanase activity in the endosperm during dormancy breaking under field conditions. Activity was initially localized to the micropylar endosperm in September and gradually spread throughout the entire endosperm by November (Figure 4). In November, high enzyme activity was detected in the lateral endosperm, while the micropylar region showed low activity. By March, activity in both the micropylar and lateral endosperm regions increased, coinciding with radicle protrusion.

### 2.3. Structural Changes in the Endosperm

In May 2011, at the time of seed dispersal, the embryo was underdeveloped (Figure 2A and Figure 5A) and surrounded by the endosperm (Figure 5B). The micropylar endosperm consisted of smaller cells compared to the lateral endosperm and contained five to six cell layers in front of the radicle tip. By 6 October 2011, significant changes were observed in the lateral endosperm. The lateral endosperm appeared eroded, with many areas showing loss of wall material, making the contours of the underlying cells clearly visible (Figure 6). Despite this extensive degradation in the lateral endosperm, the micropylar endosperm showed no visible signs of collapse or damage. In March 2012, a micropylar endosperm rupture occurred, leading to the completion of germination (Figure 2F).

## 3. Discussion

Endo-β-mannanase has been extensively studied in relation to seed germination across many species. However, most of these studies involve seeds with fully developed embryos that lack morphological dormancy (MD) or morphophysiological dormancy (MPD). Additionally, seeds with non-deep physiological dormancy (PD) experience relatively short dormant periods compared to seeds with deep PD. Based on fossils of gymnosperms and angiosperms, the seeds of ancestors had an underdeveloped embryo, such as those with MD and/or MPD [28]. As the embryo size increased, this likely led to the evolution of seeds with fully developed embryos that exhibited no dormancy and/or PD [17]. The seed dormancy of *Jeffersonia dubia* has been classified as deep simple MPD [22]. The requirements for germination of seeds with deep simple MPD included three steps: (1) a warm stratification period to break the PD so that underdeveloped embryos could start to grow, (2) a period of moderate temperatures during which embryo elongation occurred and was completed, and (3) a period of cold stratification during which the fully developed embryos were ready to let the radicle emerge [17]. The present results align with previous studies [22], showing that embryo elongation and germination in *J. dubia* follow this pattern (Figure 1 and Figure 2).

Previous studies observing endo-β-mannanase activity and seed germination have shown that its activity typically occurs either just before or after seed germination. For example, in the seeds of carrot and tomato, endo-β-mannanase activity increases in the micropylar endosperm before germination [7,13]. However, in seeds like lettuce, celery, date palm, and pepper, enzyme activity increases only after germination [4,10,29]. However, in *J. dubia*, a distinct pattern of endo-β-mannanase activity was observed. The first occurrence of endo-β-mannanase activity coincided with the initial elongation of the embryo (October–November) but decreased after the embryo had fully elongated (Figure 3 and Figure 4). During winter (December–February), both embryo elongation and enzyme activity were low, indicating a state of quiescence. Subsequently, a second phase of endo-β-mannanase activity occurred, triggering germination. In early March, enzyme activity increased again, aligning with germination.

The seasonal fluctuation in endo-β-mannanase activity observed during the germination process of *J. dubia* is considered a characteristic of seeds with underdeveloped embryos. However, not all seeds with underdeveloped embryos exhibit such fluctuations in endo-β-mannanase activity. Seed dormancy types associated with seeds containing underdeveloped embryos include MD and MPD. Carrot (*Daucus carota*) seeds have an MD dormancy type, where embryo elongation and germination are completed within a month without requiring specific dormancy-breaking conditions [17]. Carrot seeds exhibit an increase in endo-β-mannanase activity alongside embryo elongation, which leads to germination [30]. Another plant species with MPD dormancy is *Annona crassiflora*. This species has non-deep simple MPD, requiring cold stratification to break physiological dormancy and allow embryo elongation and germination [17]. In *A. crassiflora*, endo-β-mannanase activity remains absent during the cold stratification period but gradually increases after dormancy is broken, coinciding with embryo elongation and directly leading to germination [31]. In contrast, in *J. dubia*, endo-β-mannanase activity occurred during embryo elongation but did not directly induce germination. *Jeffersonia dubia* required a second physiological dormancy-breaking after embryo elongation. This distinction likely explains the unique fluctuations in endo-β-mannanase activity observed in this species. The phenology and temperature requirements for embryo elongation and dormancy break in *J. dubia* are similar to other species with deep simple MPD, such as *J. diphylla* (*Berberidaceae*), *Panax ginseng* (*Araliaceae*), *Cardiocrinum cordatum* (*Liliaceae*), and *Cephalotaxus wilsoniana* (*Cephalotaxaceae*), despite differences in seed maturation timing [32]. Among the nine types of MPD seed dormancy, those requiring two treatments (warm and cold stratification) to break dormancy, like *J. dubia*, include intermediate simple MPD, deep simple MPD, non-deep epicotyl simple MPD, deep epicotyl simple MPD, and non-deep complex MPD [17]. Although studies on endo-β-mannanase activity in these seed dormancy types have not yet been conducted, they may exhibit similar seasonal fluctuations as observed in *J. dubia*.

At seed dispersal in May, the embryos of *J. dubia* were underdeveloped and remained dormant for three months (June–September), with no detectable endo-β-mannanase activity in the micropylar or lateral endosperm (Figure 3 and Figure 4). Enzyme activity first appeared in the micropylar endosperm region in September, coinciding with the onset of embryo elongation. It is worth noting that this activity may not be exclusively from the micropylar endosperm, as the preparation method could include parts of the lateral endosperm. Similar observations have been made in seeds with underdeveloped embryos, such as carrot and *A. crassiflora*, where endo-β-mannanase activity in the micropylar endosperm coincides with the beginning of embryo elongation [30,31]. The cell wall contains significant amounts of galactomannan, which is a source of stored reserves [3]. Endo-β-mannanase plays a crucial role in germination by hydrolyzing galactomannan in the endosperm wall. In some species, such as rice and date palm, endo-β-mannanase activity supports the mobilization of galactomannan reserves for early seedling growth [33,34]. In this experiment, endo-β-mannanase activity around the micropylar endosperm, including part of the lateral endosperm surrounding the embryo, led to cell breakdown around the embryo in *J. dubia* and the formation of a cavity (Figure 6). It may provide a carbohydrate source for early embryo development. Genes encoding endo-β-mannanase expressed in the lateral and micropylar endosperm are different. In tomato seeds, three genes encoding MANs (*LeMAN* 1, 2, and 3) are expressed in different parts of the endosperm [35]. *LeMAN1* and *LeMAN3* were expressed in the lateral endosperm, where they participated in the mobilization of mannan reserves within the cell wall [36,37], while *LeMAN2* was expressed in the micropylar endosperm and involved in softening the cell wall to allow radicle emergence [37,38]. In *J. dubia*, endo-β-mannanase increased during the embryo elongation period following warm stratification (Figure 3 and Figure 4) and was likely expressed in the lateral endosperm to mobilize stored reserves. However, after cold stratification, the endo-β-mannanase expressed in March was produced in the micropylar endosperm, alongside the lateral endosperm, contributing to endosperm weakening and facilitating radicle protrusion. In the seeds with MPD dormancy, the micropylar endosperm weakened just before germination. For example, ash (*Fraxinus excelsior*) has the same deep simple MPD dormancy as *J. dubia* [18,39]. Warm stratification breaks the dormancy in its small, fully differentiated embryos, while cold stratification enables germination. During stratification, changes in endosperm properties, such as reduced puncture force, suggest weakening of the endosperm layer. SEM observations in *J. dubia* showed that endo-β-mannanase coincided with pronounced erosion of the endosperm cell wall (Figure 5 and Figure 6). This enzyme likely facilitates endosperm weakening by breaking intercellular adhesion during warm and cold stratification.

In summary, *J. dubia* seeds with both morphological and physiological dormancy require extended periods of warm and cold stratification for germination. The increase in endo-β-mannanase activity coincided with embryo elongation, decreased during winter quiescence, and resumed in spring at germination. This study is the first to reveal the seasonal pattern of endo-β-mannanase activity in seeds with deep MPD dormancy, providing valuable insights into the germination mechanisms of basal angiosperms.

## 4. Materials and Methods

### 4.1. Seeds

Seed capsules of *J. dubia*, each containing 10–15 seeds, were harvested from a population in Yongin (37°05′ N, 127°24′ E), Gyeonggi Province, South Korea, on 22 May 2011. This population grows in a mesic deciduous woodland within the Hantaek Botanical Garden, an ex situ conservation habitat. The seeds were manually extracted from the capsules, winnowed, and air-dried at room temperature (22–25 °C) for four days. The dried seeds were then sealed in impermeable plastic bags and stored at 5 °C for another four days before the experiments commenced.

### 4.2. Embryo Elongation and Germination in a Field Condition

Embryo elongation in *J. dubia* seeds was monitored under field conditions by burying the seeds and exhuming them at predetermined intervals. Ten nylon bags, each containing 20 seeds and 10 g of white sand, were buried at a depth of 3 cm in an experimental garden on the campus of Seoul National University, Seoul, Republic of Korea, on 30 May 2011. Soil temperature at a burial depth of 3 cm was recorded every 30 min using a thermo data logger (Watch Dog Model 450, Spectrum Technologies, Inc., Plainfield, IL, USA), and weekly maximum and minimum temperatures were calculated. On the 6th day of each month, 10 seeds were randomly removed from one nylon bag, and embryo lengths were measured. The seeds were bisected under a dissecting microscope using a razor blade, and embryo lengths were recorded with an ocular micrometer from 6 June 2011 to 6 February 2012. The ratio of embryo length to seed length (E:S ratio) was calculated. Photographs of the seeds were captured using a digital microscope (MV200UV, Cosview Technologies Co., Shenzhen, China).

The timing of seed germination under field conditions was determined using three nylon bags, each containing 100 seeds, buried at a depth of approximately 1 cm in the same experimental garden. Seeds with radicle emergence exceeding 2 mm were counted and removed weekly from May 2011 to March 2012. Germination was defined as the protrusion and elongation of the radicle. Intact seeds that did not germinate were reburied for continued monitoring.

### 4.3. Endo-β-Mannanase Extraction and Assay

Ten *J. dubia* seeds collected monthly under field conditions from 6 June 2011 to 6 March 2012 were used to measure endo-β-mannanase activity in the endosperm. Each seed was dissected into two parts: the micropylar endosperm region (approximately one-fifth of the seed length) and the lateral endosperm region (approximately four-fifths of the seed length). The endosperm tissues were homogenized in 1.0 mL of chilled 0.2 M K-phosphate buffer (pH 6.8) using a mortar and pestle. The homogenate was centrifuged at 10,000× *g* for 5 min, and the supernatant was used for the enzyme activity assay.

Enzyme activity was determined using an activity gel (8 mm thickness) containing 0.5% (*w*/*v*) locust bean gum (Sigma, St Louis, MO, USA), McIlvaine buffer (0.05 M citric acid/0.1 M Na_2_HPO_4_, pH 5.0), and 0.8% Phytogel (Sigma), prepared on Petri plates. Ten microliters of the extract were pipetted into 2 mm diameter wells punched into the gel. The gel was incubated at 40 °C for 16 h in the dark, washed with McIlvaine buffer (pH 5.0) for 30 min, stained with 0.5% (*w*/*v*) Congo Red (Sigma) for 15 min, and rinsed in 96% ethanol for 10 min. It was then destained with 1 M NaCl for 6 h. All staining and destaining steps were performed on a rotating platform. A standard curve was generated using commercial endo-β-mannanase from *Aspergillus niger* (Megazyme, North Rocks, Sydney, Australia), and calculations were performed following the method of Downie et al. [40].

For localization of endo-β-mannanase activity, 15 seeds collected monthly from field conditions were used. The seed coats were removed with tweezers, and the seeds were rinsed in deionized water before being bisected with a razor blade. The cut seed parts were blotted dry on filter paper and placed cut side down on an activity gel (2 mm thick) prepared as described above. The gels were incubated at room temperature for 1 h, after which the seed parts were removed. The gels were subsequently stained and destained using the same protocol as for the enzyme activity assay.

### 4.4. Structural Changes in the Endosperm

The micropylar and lateral endosperms of *J. dubia* seeds were analyzed using a scanning electron microscope (SEM). Ten seeds were exhumed on 22 May and 6 October 2011, respectively, and fixed in 2.5% formaldehyde prepared in 0.05 M sodium cacodylate buffer at 5 °C. The seeds were then longitudinally sectioned using a razor blade to expose the endosperm regions. The sections were examined under a field-emission scanning electron microscope (SUPRA 55VP, Carl Zeiss, Wetzlar, Germany) operated at an acceleration voltage of 15 kV.

## Figures and Tables

**Figure 1 plants-14-00251-f001:**
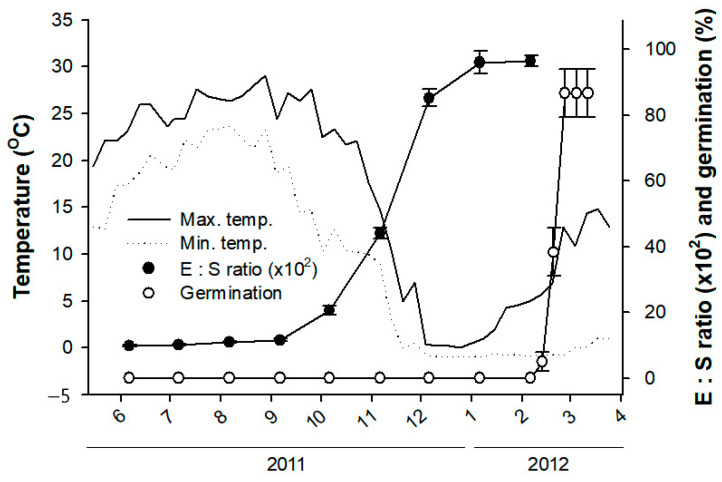
Mean weekly minimum and maximum soil temperatures, phenology of embryo elongation, and germination in *Jeffersonia dubia* seeds sown under natural conditions at a depth of 3 cm on 30 May 2011. The E:S ratio refers to the average length of the embryo relative to the seed.

**Figure 2 plants-14-00251-f002:**
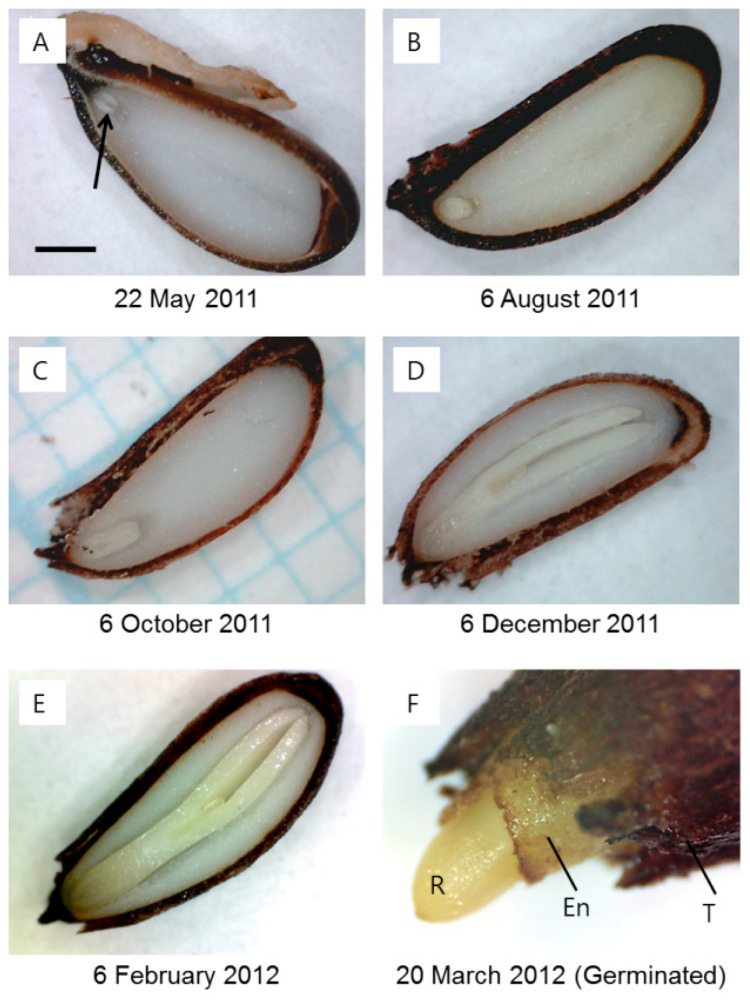
Embryo (arrow) stages of development in seeds of *Jeffersonia dubia* outdoors in Seoul, Korea. Embryos showed the heart-shaped stage at the time of dispersal on 22 May (**A**) and 6 August (**B**), the torpedo stage on 6 October (**C**), and the cotyledon stage on 6 December (**D**) and 6 February of the following year (**E**), and the radicle protrusion through the micropylar endosperm (**F**). R: radicle; En: endosperm; T: testa.

**Figure 3 plants-14-00251-f003:**
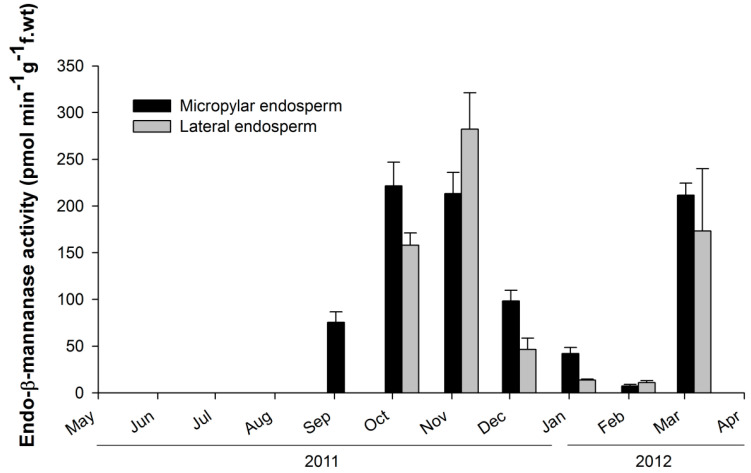
Endo-β-mannanase activity in the micropylar and lateral endosperm of *Jeffersonia dubia* seeds under field conditions from June 2011 to March 2012. Bars are ±1 SE.

**Figure 4 plants-14-00251-f004:**
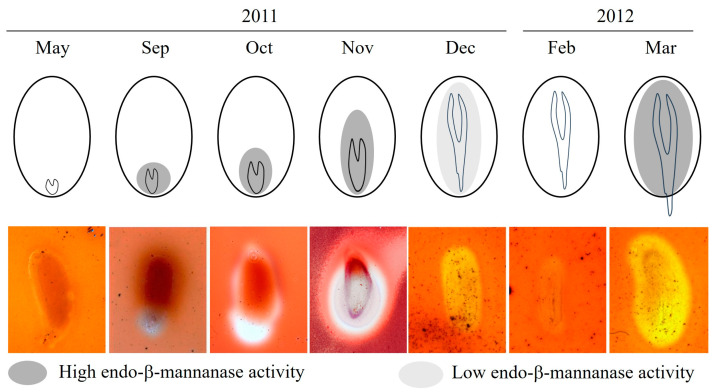
Tissue printing of longitudinally cut seeds during dormancy breaking, showing endo-β-mannanase activity as clearings in micropylar and lateral endosperm in May, September, October, November, and December 2011, and in February and March 2012.

**Figure 5 plants-14-00251-f005:**
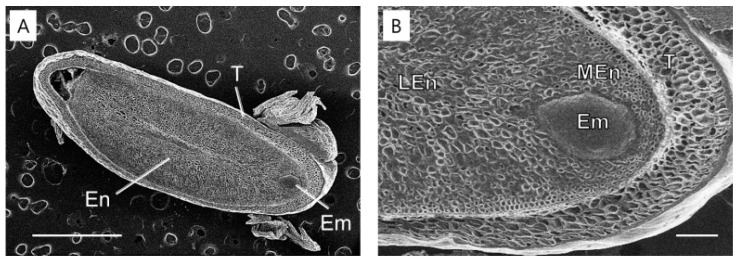
Scanning electron microscopy of *Jeffersonia dubia* seeds at dispersal time on 22 May 2011. (**A**) General view of the embryo (Em), endosperm (En), and testa (T); (**B**) Cells of the lateral endosperm (LEn), micropylar endosperm (MEn), and testa. Bars indicate 1 mm for (**A**) and 200 μm for (**B**).

**Figure 6 plants-14-00251-f006:**
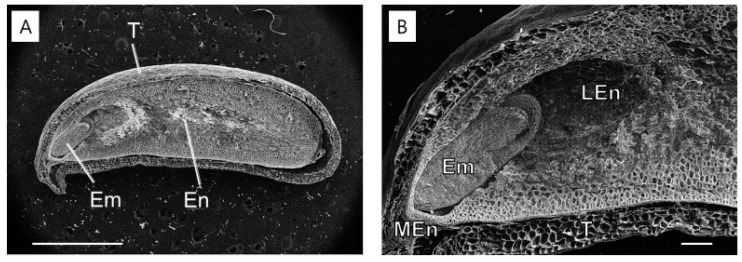
Scanning electron microscopy of *Jeffersonia dubia* seeds with the developing embryo exposed to natural field conditions until 6 October 2011. (**A**) General view of the embryo (Em), endosperm (En), and testa (T); (**B**) Cells of the lateral endosperm (LEn), micropylar endosperm (MEn), and testa. Bars indicate 1 mm for (**A**) and 200 μm for (**B**).

## Data Availability

Data are contained within the article.

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
