# Peer review of "Seasonal Pattern of Endo-β-Mannanase Activity During Germination of Jeffersonia dubia, Exhibiting Morphophysiological Dormancy"

_plants, 2025, doi:10.3390/plants14020251_

Round 1
Reviewer 1 Report
Comments and Suggestions for Authors
Dear authors, The work is very interesting, especially since it explores deep morphophysiological seed dormancy mechanisms. Details of the review are included in the attached PDF, but I would like to highlight the main reservations:
1. Introduction—The plant, its life cycle, and ecological adaptation related to the occurrence of MPD are not described.
2. In the Discussion, the topic of enzyme action mechanisms in relation to germination stages is not explored. In my opinion, the inference based on comparing the germination of seeds, e.g. carrots (which do not have MPD) and the observed species Jeffersonia dubia in the context of the activity of the Endo-B-mannanase enzyme is too superficial. Yes, such a comparison could be made, but one cannot compare different stages, e.g. germination of carrot seeds and stratification of Jeffersonia dubia seeds.
3. What is missing in Results—there is no description of the developmental stages of the embryo. All processes of seed maturation (embryo elongation) are called growth or germination, and surprisingly, sometimes, those words are used as synonyms. That is an overinterpretation.
This could be equalled and corrected for scientific soundness according to botanical nomenclature.

Author Response
Comments 1: Introduction—The plant, its life cycle, and ecological adaptation related to the occurrence of MPD are not described.
Response 1: Thank you very much for taking the time to provide such valuable feedback. Based on your comments, I have made the following revisions. I have added the ecological significance of MPD as follows (Line 66-76):
“Seeds with MPD do not germinate even under optimal conditions for germination and sometimes require an exceptionally long dormancy-breaking period. This is considered an advanced survival strategy for the plant [15]. For example, plant species with the deep simple MPD, one of the nine MPD types, include Panax ginseng [17], Sambucus canadensis [18], and two species of the genus Jeffersonia [19,20], among others. These plant seeds re-quire experiencing two seasons—summer and winter—before germinating in the follow-ing spring. These plants bloom in early spring, and their seeds mature before summer. Their germination strategy might avoid competition with other plants dominant in sum-mer, prevents exposure to dry summer conditions, and seeks the most favorable season for germination, which is spring, avoiding both the harsh conditions of summer and the cold winter periods.”
Comments 2: In the Discussion, the topic of enzyme action mechanisms in relation to germination stages is not explored. In my opinion, the inference based on comparing the germination of seeds, e.g. carrots (which do not have MPD) and the observed species Jeffersonia dubia in the context of the activity of the Endo-B-mannanase enzyme is too superficial. Yes, such a comparison could be made, but one cannot compare different stages, e.g. germination of carrot seeds and stratification of Jeffersonia dubia seeds.
Response 2: Thank you for the insightful suggestion. I have elaborated on the role and action mechanisms of endo-β-mannanase in seed germination in the discussion (Line 222-236). Moreover, instead of a simple comparison across species, I have focused more on the differences in endo-β-mannanase expression according to the types of seed dormancy (Line 187-203).
(Line 222-236) “The cell wall contains significant amounts of galactomannan, which are a source of stored reserves [3]. Endo-β-mannanase plays a crucial role in germination by hydrolyzing galactomannan in the endosperm wall. In some species, such as rice and date palm, endo-β-mannanase activity supports the mobilization of galactomannan reserves for early seedling growth [24,25]. In this experiment, endo-β-mannanase activity around the micropylar endosperm, including part of the lateral endosperm surrounding the embryo, led to cell breakdown around the embryo in J. dubia and the formation of a cavity (Figure 6). It may provide a carbohydrate source for early embryo development. Genes encoding endo-β-mannanase expressed in the lateral and micropylar endosperm are different. In tomato seeds, three genes encoding MANs (LeMAN 1, 2, and 3) are expressed in different parts of endosperm [26]. LeMAN1 and LeMAN3 were expressed in the lateral endosperm, where they participated in the mobilization of mannan reserves within the cell wall [27,28], while LeMAN2 was expressed in the micropylar endosperm and involved in softening the cell wall to allow the radicle emergence [28,29].”
(Line 187-203) “The seasonal fluctuation in endo-β-mannanase activity observed during the germination process of J. dubia is considered a characteristic of seeds with underdeveloped embryos. However, not all seeds with underdeveloped embryos exhibit such fluctuations in endo-β-mannanase activity. Seed dormancy types associated with seeds containing underdeveloped embryos include MD and MPD. Carrot (Daucus carota) seeds have an MD dormancy type, where embryo growth and germination are completed within a month without requiring specific dormancy-breaking conditions [14]. Carrot seeds exhibit an increase in endo-β-mannanase activity alongside embryo growth, which leads to germination [21]. Another plant species with MPD dormancy is Annona crassiflora. This species has non-deep simple MPD, requiring cold stratification to break physiological dormancy and allow embryo growth and germination [14]. In A. crassiflora, endo-β-mannanase activity remains inactive during the cold stratification period but gradually increases after dormancy is broken, coinciding with embryo growth and directly leading to germination [22]. In contrast, in J. dubia, endo-β-mannanase activity occurred during embryo growth but did not directly induce germination. Jeffersonia dubia required a second physiological dormancy-breaking after embryo growth. This distinction likely explains the unique fluctuations in endo-β-mannanase activity observed in this species.”
Comments 3: What is missing in Results—there is no description of the developmental stages of the embryo. All processes of seed maturation (embryo elongation) are called growth or germination, and surprisingly, sometimes, those words are used as synonyms. That is an overinterpretation. This could be equalled and corrected for scientific soundness according to botanical nomenclature.
Response 3: The seeds of Jeffersonia dubia contained underdeveloped, heart-shaped embryos of freshly matured seeds, which elongated during the dormancy-breaking period. We also have described in the manuscript the development of the embryo in J. dubia seeds, progressing through the heart, torpedo, and cotyledon-shaped stages. We added more information about this to the results (Line 87-93). To avoid confusion, all instances of "embryo growth" in the manuscript have been revised to "embryo elongation."
(Line 87-93) “The seeds of J. dubia had underdeveloped embryos with a heart shape of freshly matured seeds and the average embryo-to-seed (E : S) ratio (×10²) was only 9.0 ± 0.2 (mean ± SE) (Figure 1, 2A). Significant embryo elongation began between 6 September and 6 October 2011 (Figure 1). During this period, the embryo could be observed developing into a torpedo shape (Figure 2B–2C). Embryo developed into the cotyledon stage between 6 October and 6 December 2011 (Figure 2D), coinciding with mean maximum and minimum temperatures of 20°C and 0°C, respectively (Figure 1).”
Thank you also for the PDF file you attached. We have made sure to incorporate as many of the revisions as possible into the updated manuscript.
Reviewer 2 Report
Comments and Suggestions for Authors
Overall, this research paper presents a valuable contribution to the understanding of seed dormancy mechanisms in species with morphophysiological dormancy, despite some limitations in scope and methodology.
In my opinion strength of this research:
1. Long-term study: The research was conducted over a full year, allowing for a detailed examination of seasonal patterns in enzyme activity and embryo growth.
2. Clear data presentation: The study includes well-designed figures that effectively illustrate the temporal and spatial patterns of enzyme activity and embryo growth.
Weakness of the study:
1. Given that MPD is considered one of the most primitive dormancy classes, a more extensive literature review and discussion on the evolutionary implications of the findings could have enhanced the paper's impact.
2. The study primarily focuses on physiological and morphological aspects. Including molecular analyses of gene expression related to endo-β-mannanase production could have provided additional insights into the regulatory mechanisms of dormancy breaking.
The manuscript is very well-written with very few noticeable errors. However, there are a couple of minor issues:
Line 38: Please use regular font instead of bold for keywords
Figure 5 Line 132-134 should be in regular font not bold
In line 151: "was ready to germinate" should be "were ready to germinate" to match the plural subject "embryos".
Line 213. Please be consistence with the format of the subheadings for 4.2.
Line 259. should be 4.4.

Author Response
Comments 1: Given that MPD is considered one of the most primitive dormancy classes, a more extensive literature review and discussion on the evolutionary implications of the findings could have enhanced the paper's impact.
Response 1: Thank you for the valuable comment. We added more references regarding endo-β-mannanase expression in seeds with MD and MPD dormancy types to the discussion (Line 187-213).
Comments 2: The study primarily focuses on physiological and morphological aspects. Including molecular analyses of gene expression related to endo-β-mannanase production could have provided additional insights into the regulatory mechanisms of dormancy breaking.
Response 2: We haven't examined endo-β-mannanase gene expression, but we added a discussion in the manuscript about the discovery of genes expressed in the lateral and micropylar endosperm during the germination process in lettuce (Line 230-236).
Comments 3: Line 38: Please use regular font instead of bold for keywords
Response 3: We revised.
Comments 4: Figure 5 Line 132-134 should be in regular font not bold
Response 4: We revised.
Comments 5: In line 151: "was ready to germinate" should be "were ready to germinate" to match the plural subject "embryos".
Response 5: We revised.
Comments 6: Line 213. Please be consistence with the format of the subheadings for 4.2.
Response 6: We revised.
Comments 7: Line 259. should be 4.4.
Response 7: We revised.
Reviewer 3 Report
Comments and Suggestions for Authors
The manuscript describes a study of seed germination dynamics and activity of enzymes that enable endosperm rupture in seeds with morphophysiological dormancy. Although the activity of endo beta mannanase has been studied in different species, this topic is important for understanding the physiology of germination in seeds with MPD.
The manuscript is generally well-written. In the Introduction section, more information could be provided on the species (Jeffersonia dubia). It is stated that this species is a basal angiosperm, so perhaps a brief statement on the phylogenetic relationship of this species can be added. The presentation of the results and the description of the methods are clear. The discussion cites relevant studies by other authors
Author Response
Comments 1: The manuscript describes a study of seed germination dynamics and activity of enzymes that enable endosperm rupture in seeds with morphophysiological dormancy. Although the activity of endo beta mannanase has been studied in different species, this topic is important for understanding the physiology of germination in seeds with MPD.
The manuscript is generally well-written. In the Introduction section, more information could be provided on the species (Jeffersonia dubia). It is stated that this species is a basal angiosperm, so perhaps a brief statement on the phylogenetic relationship of this species can be added. The presentation of the results and the description of the methods are clear. The discussion cites relevant studies by other authors
Response 1: Thank you for the valuable comment. We added brief statement on the phylogenetic relationship of seed dormancy type (Line 163-166).
(Line 163-166) “Based on fossil of gymnosperms and angiosperms, seeds of ancestors had an underdeveloped embryo, such as those with MD and/or MPD [20]. As the embryo size increased, this likely led to the evolution of seeds with fully developed embryos that exhibited no dormancy and/or PD [14].”
Reviewer 4 Report
Comments and Suggestions for Authors
Dear Authors
The paper examines the Endo-β-mannanase seasonal pattern during germination of Jeffersonia dubia (the species should be mentioned in the title)
The manuscript is well writing and presents data to deepen the understanding In basal angiosperms.
The topic is interesting and the presentation of the data is OK.
There are representative figures of the experimentation.
The presentation is quite clear and concise.
The manuscript is adequate with the scope of 'Plants'
However, there are two main points:
1) The data are coming from 2011
2) There are a few data only for seasonal pattern of Endo-β-mannanase
3) There are only 23 references (from 1983-2015)
I think that the authors should enrice the discussion comparing the data of the present research with previous research on in vitro propagation of the species.
I think also that there is a number of ecophysiology studies on J. dubia and other studies on species with MPD that could be discussed.
The authors could also run again their study and present comparatively the data of two different collections. A correlation of these data could be very interesting..
Author Response
Comments 1: The paper examines the Endo-β-mannanase seasonal pattern during germination of Jeffersonia dubia (the species should be mentioned in the title)
Response 1: We revised title. “Seasonal Pattern of Endo-b-mannanase Activity during Ger-mination of Jeffersonia dubia, Exhibiting Morphophysiological Dormancy”
Comments 2: The manuscript is adequate with the scope of 'Plants'. However, there are two main points:
- The data are coming from 2011
- There are a few data only for seasonal pattern of Endo-β-mannanase
- There are only 23 references (from 1983-2015)
Response 2:
- Although this study was conducted in 2011, research on endo-β-mannanase activity in MPD, which belongs to the ancestor seed dormancy type, has not been conducted to date. Therefore, it is considered to have significant academic value.
- Since the endo-β-mannanase activity was monitored over the course of one year during the dormancy break of Jeffersonia dubia seeds, we thought that the data is considered substantial. Additionally, not only was the activity of endo-β-mannanase observed, but changes in the endosperm cells were also examined using SEM.
- We have added references, bringing the total to 33, and included recent studies, with the years ranging from 1983 to 2022.
Comments 3: I think that the authors should enrice the discussion comparing the data of the present research with previous research on in vitro propagation of the species. I think also that there is a number of ecophysiology studies on J. dubia and other studies on species with MPD that could be discussed.
Response 3: We have further discussed the endo-β-mannanase activity in other plant species (Line 187-212, Line 222-243).
Comments 4: The authors could also run again their study and present comparatively the data of two different collections. A correlation of these data could be very interesting.
Response 4: Although additional experiments were not conducted on different collections, We have added a statement speculating on the potential endo-β-mannanase activity in plants with different seed types (Line 207-213).
Round 2
Reviewer 1 Report
Comments and Suggestions for Authors
Dear Authors,
Thanks for adjusting the manuscript according to the review. I found a few other simple mistakes or lack of information to correct. For details, see the attached PDF.
The Literature cited in the publication is quite old. I suggest adding some publications about seed dormancy and its regulation published after 2020 to update knowledge and improve the quality of the manuscript. Many such publications have been published, even in the Plants Journal.
Kind regards

Author Response
Comments 1: The Literature cited in the publication is quite old. I suggest adding some publications about seed dormancy and its regulation published after 2020 to update knowledge and improve the quality of the manuscript. Many such publications have been published, even in the Plants Journal.
Response 1: Thank you for your comments. We have added four papers published after 2021 to the manuscript (Ref. 8, 14, 23, 24).
Comments 2: (Line 86) Can you provide more informations about fruit and seeds of J. dubia? What is the type of the fruit? that are the dimantions of the seeds? their weight? color? How they are dispersed in nature?
Response 2: We added information on the seed characteristics of J. dubia and its distribution characteristics in nature. (Line 88-93)
Comments 3: (Line 108) soil at à in natural conditions
Response 3: We changed. (Line 114)
Comments 4: (Line 115) what is seen on F?
Response 4: We have described (F). (Line 121)
Comments 5: (Line 117) Observation of ....
Response 5: We changed. (Line 124)
Comments 6: (Line 120) (what does it mean exatly? how small it was?
Response 6: We added the E:S ratio. (Line 126)
Comments 7: (Line 130) there is no "D" on Figure 4
Response 7: Thank you for your careful review of the manuscript. We deleted.
Comments 8: (Line 132) there is no "G" on Figure 4
Response 8: We deleted.
Reviewer 4 Report
Comments and Suggestions for Authors
Dear Authors and Editors
The paper examines the Endo-β-mannanase seasonal pattern during germination of Jeffersonia dubia
The manuscript is well writing and presents data to deepen the understanding In basal angiosperms.
The topic is interesting and the presentation of the data is OK.
There are representative figures of the experimentation.
The presentation is quite clear and concise.
The manuscript is adequate with the scope of 'Plants'
In my previous report I was mentioned that the data are coming from 2011, there are a few data only for seasonal pattern of Endo-β-mannanase and there were only 23 references (from 1983-2015)
I suggested that the authors should enrice the discussion comparing their findings to previous research on the species. More studies on species with MPD could be discussed.
Finally the authors could run again their study and present comparatively the data of two different collections.
Regarding the present improved manuscript:
Title
The title improved and concludes the scientific name of the species. It is specific and relevant.
Keywords
They are specific to the article. They should be placed alphabetically. On the other hand Berberidaceae is mentioned in the keywords but in the text it is not clear that the species belongs to Berberidaceae.
Absrtact
Introduction
The introduction highlights the background and its importance.
It defines the purpose of the work and its significance, including specific hypotheses being tested. In the improved edition there are key publications cited.
It mentions the main aim of the work and highlights the main conclusions.
Still, a question there is: Is there any other special characteristic of the present species? Is this medicinal, ornamental, or other? I think you could include information of special characteristic of the species (if there is any special characteristic it could reveal more the significance of the present study)
Materials and Methods
They well describe used methods and protocols.
Results
The revised paper includes an improved figure (Fig 4) showing endo-mannanase activity.
All figures are representative.
A concise and precise description of the experimental results is provided.
Discussion
This section was the weakest in the manuscript (MS). In the improved MS the authors discuss the results and previous studies and of the working hypotheses. They disscusse the endo-β-mannanase activity in other plant species
The authos have added references including recent studies (1983-2022).
A statement speculating on the potential endo-β-mannanase activity in plants with different seed types is now including (Line 207-213).
Conclusions
The section is short and clear.
The present well improved MS could be published in the Plants after minor revision
There are 2-3 issues, blue highlighted, in the submitted .pdf.

Author Response
Comments 1: Keywords
They are specific to the article. They should be placed alphabetically. On the other hand Berberidaceae is mentioned in the keywords but in the text it is not clear that the species belongs to Berberidaceae.
Response 1: Regarding the genus (Berberidaceae), it has been removed from the keywords.
Comments 2: Introduction
Still, a question there is: Is there any other special characteristic of the present species? Is this medicinal, ornamental, or other? I think you could include information of special characteristic of the species (if there is any special characteristic it could reveal more the significance of the present study)
Response 2: Thank you very much for taking the time to provide such valuable feedback. The plant J. dubia is used as an ornamental plant in East Asia. Among its extracts is berberine, a compound known for its significant pharmacological effects. We have included additional information and relevant literature on this topic. (Line 80-84)
Comments 3: Please write the family and the syn:
According to the Plant List the species is named Plagiorhegma dubium Maxim.
You could mention to that It is important for someone who is researching on the species https://wfoplantlist.org/taxon/wfo-0001146550-2022-12?page=1
Response 3: Thank you very much. I was unaware that the scientific name Jefferdonia dubia had been revised. Thank you for clarifying. I have now updated it to reflect that Jefferdonia dubia is a synonym of Plagiorhegma dubium (Line 81). And I added the accepted name Plagiorhegma dubium to the keywords (Line 38).